


# Wintertime aerosol dominated by solid fuel burning emissions across Ireland: insight into the spatial and chemical variation of submicron aerosol

Chunshui Lin[1,2,3], Darius Ceburnis[1], Ru-Jin Huang[1,2,3*], Wei Xu[1,2], Teresa Spohn[1], Damien Martin[1], Paul Buckley[4], John Wenger[4], Stig Hellebust[4], Matteo Rinaldi[5], Maria Cristina Facchini[5], Colin O'Dowd[1*], and Jurgita Ovadnevaite[1]

[1]School of Physics, Ryan Institute's Centre for Climate and & Pollution Studies, and Marine Renewable Energy Ireland, National University of Ireland Galway. University Road, Galway. H91 CF50, Ireland
[2]State Key Laboratory of Loess and Quaternary Geology and Key Laboratory of Aerosol Chemistry and Physics, Chinese Academy of Sciences, 710061, Xi'an, China
[3]Center for Excellence in Quaternary Science and Global Change, Institute of Earth Environment, Chinese Academy of Sciences, Xi'an 710061, China
[4]School of Chemistry and Environmental Research Institute, University College Cork, Cork, Ireland
[5]Istituto di Scienze dell'Atmosfera e del Clima, Consiglio Nazionale delle Ricerche, 40129, Bologna, Italy

*Correspondence to*: Ru-Jin Huang (rujin.huang@ieecas.cn) and Colin O'Dowd (colin.odowd@nuigalway.ie)

**Abstract.** To get an insight into the spatial and chemical variation of the submicron aerosol, a nationwide characterization of wintertime $PM_1$ was performed using an Aerosol Chemical Speciation Monitor (ACSM) and Aethalometer at four representative sites across Ireland. Dublin, the capital city of Ireland, was the most polluted area with an average $PM_1$ concentration of 8.6 μg m$^{-3}$, ranging from <0.5 μg m$^{-3}$ to 146.8 μg m$^{-3}$ in December 2016. The $PM_1$ in Dublin was mainly composed of carbonaceous aerosol (organic aerosol (OA) + black carbon (BC)) which, on average, accounted for 80% of total $PM_1$ mass during the monitoring period. Birr, a small town in the midlands area of Ireland with a population <1% of that in Dublin, had comparable $PM_1$ concentrations with an average of 4.8 μg m$^{-3}$, ranging from <0.5 to 63.0 μg m$^{-3}$ in December 2015. Similarly, the $PM_1$ in Birr was also mainly composed of carbonaceous aerosol, accounting for 77% of total $PM_1$ mass. OA source apportionment results show that local emissions from residential heating were the dominant contributors (65-74% of the OA) at the two sites, with solid fuel burning, on average, contributing 48-50% of the total OA. On the other hand, Carnsore Point and Mace Head, which are both regional background coastal sites, showed lower average $PM_1$ concentrations (2.2 μg m$^{-3}$ for Carnsore Point in December 2016 and 0.7 μg m$^{-3}$ for Mace Head in January 2013) due to the distance from emission sources. Both sites were dominated by secondary aerosol comprising oxygenated OA (OOA), nitrate, sulfate, and ammonium. This nationwide source apportionment study highlights the large contribution of residential solid fuel burning to urban air pollution and identifies specific sources that should be targeted to improve air quality. On the other hand, this study also shows that rural and coastal areas are dominated by secondary aerosol from regional transport, which is more difficult to tackle. Detailed characterization of the spatial and chemical variation of submicron aerosol in this relatively less studied



Western European region have significant implications for air quality policies and mitigation strategies, as well as for regional-transport aerosol modeling.

## 1 Introduction

Atmospheric aerosol particles such as $PM_{2.5}$ (particulate matter with diameter less than 2.5 µm) have adverse effects on human
health including deterioration of the respiratory system, asthma, pulmonary disease and even premature mortality (Pope III et al., 2002; Pope III and Dockery, 2006; Sandström et al., 2005). Aerosol particles also influence the earth's radiative budget directly through absorbing and scattering sunlight, and indirectly by acting as cloud condensation nuclei (Charlson et al., 1992; O'Dowd et al., 2004; Hallquist et al., 2009; Fuzzi et al., 2015). PM is a highly complex mixture in constant evolution, emitted from various sources such as road vehicles, wood burning, and cooking. PM is also formed from the oxidation of gas-phase
precursors (e.g., $NO_x$, $SO_2$, and volatile organic compounds (VOCs)) in the atmosphere. Therefore, a better understanding of aerosol sources in a specific region or country can help inform policymakers to develop more cost-effective abatement strategies for PM.

    Ireland, located in the west of Europe, is home to ~5 million people with over 1 million people living in the capital city of Dublin (CSO, 2016). In the 1980s, Ireland experienced severe air pollution after a switch from oil to cheaper solid fuels such
as bituminous coal for domestic space and water heating (Goodman et al., 2009). In Dublin, citywide averages of the black smoke concentration exceeded 750 µg m$^{-3}$ during one particular pollution event in January 1982 (Kelly and Clancy, 1984). The Irish government subsequently introduced a ban on the marketing, sale, and distribution of bituminous coal in Dublin in 1990. The coal ban led to a 70% reduction in black smoke concentration and 10-16% reduction in respiratory and cardiovascular mortality cases over the 6 years after the ban (Clancy et al., 2002). The ban was later extended to 29 Low
Smoke Zones (www.dccae.gov.ie) across Ireland including Cork and Galway.

    In recent years, a number of studies have suggested that the ban on bituminous coal alone was not sufficient because other solid fuels such as peat and wood emit similar or higher amounts of PM when burned (Kourtchev et al., 2011; Dall'Osto et al., 2013; Lin et al., 2017; Lin et al., 2018). For example, in Cork city, Kourtchev et al., (2011) attributed ~75% of measured OC mass concentration to domestic solid fuel burning during wintertime through analyzing molecular markers on filter samples
using gas chromatography/mass spectrometry. However, filter-based studies often suffer from low time resolution and relatively large uncertainty due to the filter sampling artifacts. The introduction of near real-time monitoring of the chemical composition of PM using the Aerodyne Aerosol Mass Spectrometer (AMS) (Canagaratna et al., 2007) and Aerosol Chemical Speciation Monitor (ACSM) (Ng et al., 2011) has improved the characterization and source apportionment of PM. For example, in Cork city, Dall'Osto et al., (2013) attributed 23% of OA mass to wood burning and 21% to peat and coal burning by positive
matrix factorization (PMF) analysis of the AMS organic mass spectra. In Galway city, Lin et al., (2017) attributed up to 39% of OA to peat burning and 11% to wood burning during winter by PMF analysis of the ACSM spectra using the multilinear-engine (ME-2). In a later study in Dublin city, up to 70% of $PM_1$ was attributed to peat and wood burning during pollution



episodes (Lin et al., 2018). However, most of these studies were conducted in urban areas, and the magnitude of PM pollution and the sources of PM in rural areas remain unknown. Moreover, simultaneous measurements at both the urban and rural sites are insightful to investigate the spatial and chemical variation of the aerosol and to evaluate local and regional aerosol sources.

In this study, an ACSM and an aethalometer (AE-33) were deployed to characterize $PM_1$ chemical composition at four sites

across Ireland during wintertime. These sites include an urban background location in Dublin, a site in the small town of Birr in the midlands, and two regional background sites, one located on the east coast (Carnsore Point) and the other on the west coast (Mace Head). The chemical composition data was used to investigate the major pollution sources and assess their impact on air quality at each site (Sect. 3.1). The comparison of the simultaneous measurements conducted in Dublin and Carnsore Point was conducted to investigate the local and regional sources of $PM_1$ (Sect. 3.2). Finally, in Sect. 3.3, we compared the

chemical composition of $PM_1$ and OA source apportionment results at these four sites to provide an overview of the spatial and chemical variation of $PM_1$ across Ireland.

## 2 Experimental methods

### 2.1 Sampling sites

Four representative sites across Ireland were selected (Fig. S1). The measurement site on the campus of University College

Dublin (UCD) (53.3053° N, 6.2207° W) is an urban background site in the capital city of Dublin, Ireland. Measurements were conducted on the roof of the Science building (~30 m above the ground) at UCD. Birr is a small town which lies in the midlands area of Ireland with a population of ~5,000 and is ~150 km west of Dublin. The sampling site in Birr is located at the council yard in St. John's Place (53°05'47.1"N 7°54'29.9"W) ~100 m from the central square in the town. Mace Head Atmospheric Research Station (53° 33′N, 9° 54′W) is located on the west coast of Ireland (Jennings et al., 2003). Carnsore Point (52.19° N,

6.34° W) is located on the southeast coast of Ireland. The measurements in Dublin and Carnsore Point were conducted simultaneously in December 2016 while the campaigns in Birr and Mace Head were carried out in December 2015 and January 2013, respectively.

### 2.2 Instruments

An ACSM (Aerodyne Research Inc.) and an aethalometer (AE-33, Magee Scientific) were deployed at each site to measure

the composition and mass of submicron aerosol. The two instruments were sampling from the same $PM_{2.5}$ inlet line with isokinetic flow splitting. ACSM is a compact and low-maintenance aerosol mass spectrometer (Ng et al., 2011) employed to measure sub-micron non-refractory PM (NR-$PM_1$) with a time resolution of 30 min. A detailed description of the ACSM is given by Ng et al. (2011). Briefly, the ambient air was drawn into the cyclone with a size cut-off of 2.5 μm at a flow rate of 3 L min$^{-1}$ to remove coarse particles. The air was dried by passing through a Nafion dryer before reaching the ACSM inlet. In

the ACSM, the dried aerosol particles were focused into a narrow beam by the aerodynamic lens and entered a vacuum chamber where they were vaporized, ionized, and analyzed by a quadrupole mass spectrometer. ACSM standard data analysis software





(v 1.6.0.3) in Igor 6.37 (WaveMetrics Inc.) was utilized to process the mass concentrations of organic aerosol (OA), sulfate, nitrate, ammonium, and chloride. OA mass spectra matrix and error matrix were also extracted using this software for subsequent source apportionment studies. For all ACSM measurements, a collection efficiency (CE) of 1 was applied for all the measured species after the comparison with a collocated scanning mobility particle sizer (SMPS) (Lin et al., 2018). This

CE provided a lower limit for all ACSM-measured mass concentration. However, changes in CE did not affect the relative contribution of chemical species, since the same CE was applied to all measured species.

The aethalometers were deployed to measure black carbon (BC) at each site with a time resolution of 1 min. These instruments measure light absorption at seven wavelengths (370, 470, 520, 590, 660, 880, and 950 nm) (Drinovec et al., 2015). BC mass concentration was calculated from the change in optical attenuation at 880 nm in the selected time interval using the

mass absorption cross-section 7.77 $m^2$ $g^{-1}$ (Drinovec et al., 2015). Note that the AE-33 data at Mace Head in January 2016 were used as a reference of the BC level at Mace Head in January 2013 because AE-33 measurements were not available in January 2013. The levels of BC are expected to be similar between January 2013 and December 2016 at Mace Head as indicated by the small variation of NR-PM$_1$ components between January and December in different years at the same location (Ovadnevaite et al., 2014). The light absorption by BC particles from fossil fuel sources, e.g. traffic, is less dependent on

wavelength than those produced from biomass burning, which shows a strong increase in absorption in the near-ultraviolet and blue parts of the light spectrum (Sandradewi et al., 2008; Zotter et al., 2017). Based on this, the measured absorption coefficients at wavelengths 470 nm and 880 nm were used to attribute BC to traffic (BC$_{tr}$) and wood burning (BC$_{wb}$) sources (Sandradewi et al., 2008). Briefly, aerosol absorption coefficients ($b_{abs}$) follow the relationship $b_{abs}(\lambda_1)/b_{abs}(\lambda_2) = (\lambda_1/\lambda_2)^{-\alpha}$, where $\lambda$ is the wavelength and $\alpha$ the absorption Ångström exponent. Because the light absorption of BC$_{tr}$ is less dependent on

wavelength, traffic-related BC particles have a smaller $\alpha$ compared to wood burning-related particles. In the original aethalometer two-source model, $\alpha$ values of 1 and 2 were used for fossil fuel and biomass burning respectively (Sandradewi et al., 2008). However, the most recent evaluation recommends values of $\alpha_{tr}$=0.9 and $\alpha_{wb}$=1.68 (Zotter et al., 2017). These latter $\alpha$ values have been used here.

### 2.3 OA Source apportionment.

Positive matrix factorization (PMF) with the multilinear engine (ME-2) was utilized for OA source apportionment as with our previous studies conducted in this region (Lin et al., 2017; Lin et al., 2018; Lin et al., 2019). PMF analysis does not require *a priori* source information but often suffers from solution ambiguity resulting in a mixture of factors or inaccurate factor attributions when some factors have similar temporal variation especially at rural sites and even at urban background sites (Canonaco et al., 2013; Crippa et al., 2014; Canonaco et al., 2015). ME-2 is able to resolve more environmentally meaningful

solutions by constraining certain factors based on priori information such as factor profiles. In this study, ME-2 was conducted on the interface of SoFi 6.3 (Canonaco et al., 2013). The reference profiles of peat, coal, and wood were taken from our previous study (Lin et al., 2017) while HOA was obtained from the literature (Crippa et al., 2013). ME-2 was used to constrain the reference profiles using the *a* value approach (for example, an *a* value of 0.05 allows variability of ±5%). Sensitivity



analysis was conducted by varying *a* values (0-0.5 or 0-50% variation) to evaluate the OA factor contribution at different levels of constraint on reference factor. At the coastal sites (i.e., Mace Head and Carnsore Point), the reference sea salt profile (Ovadnevaite et al., 2012) was also included to constrain the solution (see more details in Sect. 3.1).

## 3 Results and Discussion

### 3.1 Chemical composition and sources of $PM_1$

#### 3.1.1 Dublin

Figure 1a shows the time series of $PM_1$ components measured by ACSM (i.e. OA, sulfate, nitrate, ammonium, and chloride) and AE-33 (i.e., BC) in Dublin during December 2016. The campaign-averaged $PM_1$ concentration was 8.6 µg m$^{-3}$, ranging from < 0.5 µg m$^{-3}$ to146.8 µg m$^{-3}$ (Table 1). The chemical composition of $PM_1$ was dominated by OA, which on average accounted for 57% (4.9 µg m$^{-3}$) of the total $PM_1$ mass, followed by BC, accounting for 23% (2.0 µg m$^{-3}$) of the total $PM_1$ mass. Nitrate (8% or 0.7 µg m$^{-3}$), sulfate (5% or 0.4 µg m$^{-3}$), ammonium (4% or 0.3 µg m$^{-3}$), and chloride (3% or 0.2 µg m$^{-3}$) accounted for minor fractions of $PM_1$.

Frequent pollution spikes with high OA and BC concentration (> 8.0 µg m$^{-3}$) were observed in the evening during the pollution periods (P1 - P3) while, during clean periods (C1 - C3), all $PM_1$ components were below 6.0 µg m$^{-3}$ (Figure 1). For one particulate pollution peak in the evening on 2 December 2016, the OA concentration increased up to 82.0 µg m$^{-3}$, ~17 times the OA average concentration, while BC concentration increased to 49.7 µg m$^{-3}$, ~25 times the BC average (Table 1). The simultaneous increase in both BC and OA during evening hours is a strong indication that these pollutants were emitted from a similar source, i.e. residential heating. In addition to emission sources, meteorological conditions such as wind speed and temperature were also important parameters in driving particulate air pollution. The temperature was ~1.5 times lower during the pollution periods (5.9 °C, on average) than during the clean periods (8.7 °C; Fig. S2). In addition, the wind speed was ~2.5 times lower (2.8 m s$^{-1}$ vs. 7.3 m s$^{-1}$) during pollution periods than during clean periods. These conditions commonly lead to increased air pollution in winter months due to a shallower boundary layer and less dispersion of primary emissions. The dominance of OA highlights the importance of its source apportionment to identify and quantify the major pollution sources.

To investigate the sources of the OA, unconstrained PMF (i.e., free PMF) was firstly applied to the ambient organic mass spectra. Hydrocarbon-like OA (HOA), solid fuel-burning OA (SFOA), and oxygenated OA (OOA) were identified in the free PMF runs (See Fig. S3-4 and more details in the Supplement). The free PMF solutions with higher numbers of factors provided no new meaningful factors. HOA is usually associated with traffic emission and its diurnal pattern is expected to show morning and/or evening rush hour peaks as found in other European cities e.g., in London (Allan et al., 2010) and Paris (Crippa et al., 2013). However, HOA was mixed with SFOA in this three-factor solution because the HOA profile contained a higher than expected contribution from m/z 60 (0.006) which is regarded as a marker fragment for biomass burning (BB) (Alfarra et al.,





2007). Moreover, both HOA and SFOA showed diurnal patterns with peak concentrations occurring in the evening and going into the night, indicating significant contributions from residential heating sources. Solid fuels like peat, wood, and coal have been reported to be the primary heating sources by a small proportion of households in Dublin (e.g., <5% of the household using solid fuels vs. ~95% of the households using electricity/natural gas) according to census data by Central Statistics Office (CSO, 2016). However, free PMF was not capable of separating these three types of solid fuels at the same time if they were contributing to the nighttime peaks with temporal covariation (Lin et al., 2017). Therefore, SFOA in the free PMF solution contained mixed contributions from peat, wood, and coal burning.

To reduce the mix between HOA and SFOA, and to evaluate the contribution of different types of solid fuels, the reference profiles of HOA (Crippa et al., 2013), peat, wood, and coal (Lin et al., 2017) were constrained with the *a* value approach using ME-2 (Canonaco et al., 2013). A sensitivity test by varying the *a* values from 0-0.5 with an interval of 0.1 was performed and the correlation between the resolved factors with BC measurements was evaluated and compared to choose the best solution (Fig. S5). The HOA reference profile was taken from the Paris study (Crippa et al., 2013) and a small *a* value (e.g., 0-0.2) or tight constraint was expected because the HOA profiles do not show significant variability when compared to different cities in Europe (Canonaco et al., 2013; Crippa et al., 2014). The reference profiles of peat, wood, and coal were taken from our previous study in which an ACSM was used to characterize the primary OA emissions directly from burning these fuels (Lin et al., 2017). High *a* values (e.g., 0.3-0.5) or a loose constraint led to potential mixing between these factors especially when their time series showed temporal co-variation. On the other hand, a lower *a* value (e.g., 0-0.2) reduced mixing and improved the separation by tightly constraining their individual profiles. As shown in Fig. S5, the ME-2 run with the *a* value of 0.1 was chosen as the best solution based on these criteria.

The mass spectra and time series of HOA, peat, coal, wood, and OOA are shown in Fig. 3. The HOA profile is dominated by signals at $m/z$ 27, 29, 41, 43, 55, and 57, characteristic of aliphatic hydrocarbons. Many studies have shown that HOA is usually associated with traffic emissions in urban environments (Canagaratna et al., 2004; Schneider et al., 2005; Platt et al., 2017) and the HOA/$BC_{tr}$ ratio has been reported to be an important parameter to determine the type of fuels used (e.g., diesel or gasoline) (DeWitt et al., 2015). For example, HOA/$BC_{tr}$ ratios in the range of 0.03-0.61 have been reported to be associated with diesel vehicular emission while the range of 0.9-1.7 for HOA/$BC_{tr}$ ratios is associated with gasoline vehicular emissions (DeWitt et al., 2015). In this study, the average HOA/BCtr ratio was $0.69 \pm 0.09$ during the day (8:00-15:00, local time; Fig. S6). This ratio is close to HOA/BCtr range (0.03-0.61) associated with the diesel vehicular emission and very similar to the ratio (0.61) reported in Paris (Crippa et al., 2013). Therefore, during the day, HOA was likely to be associated with diesel vehicular emissions. However, from 16:00 in the afternoon, the HOA/BCtr ratio started to increase and the ratio increased up to ~9.0 in the evening, which was significantly higher than the values associated with gasoline vehicular emissions (0.9-1.7). Therefore, HOA during the night could not be attributed to the emissions from diesel/gasoline-powered vehicles. Instead, both the diurnal of HOA and HOA/BCtr ratio indicate the HOA in the evening was mainly associated with heating sources. According to the census data from CSO (2016), natural gas, electricity, oil, wood, coal, and peat were the major types of heating sources in Dublin. Among these fuels, oil is most likely to be the source of HOA during the night because oil, gasoline,





and diesel are expected to have similar mass spectra as indicated by various ambient measurement and lab experiments (Canagaratna et al., 2004; Schneider et al., 2005; Platt et al., 2017). Assuming HOA were merely from traffic during the day and the traffic HOA/BC$_{tr}$ ratio (0.69) were stable, the traffic HOA associated with traffic during the evening (18:00-23:00) was estimated at 16% of total HOA, with the rest 84% being associated with oil heating. Over the whole period, 28% of HOA was attributed to the traffic. As shown in Fig. 2, HOA, on average, accounted for 25% of OA over the entire period. It was, therefore, estimated that traffic-related HOA accounted for 7% of the total OA and oil-related HOA accounted for 18% of the total OA.

Wood, peat, and coal are three types of solid fuels and their profiles were highly associated with their composition (Lin et al., 2017). The wood profile was characterized by higher contributions at m/z 60 (that is, *f60*) and 73 (*f73*), associated with the fragmentation of e.g. levoglucosan. *f60* was also important in the peat profile which was, however, more reduced than that in wood (0.018 in peat profile vs. 0.073 in wood profile) due to the incomplete decay of vegetation during peat formation. The coal profile featured an *f60* of nearly zero which was due to the complete decay of vegetation during coal formation. Coal is formed after millions of years under pressure and heat, resulting in rich carbon deposits. As shown in Fig. 3b, wood, peat, and coal all showed peak concentrations in the evening, corresponding to the time of residential heating activities. The solid fuel factors (the sum of wood, peat, and coal), on average, accounted for 50% of OA, highlighting its important role in driving the pollution events in Dublin (Fig. 2). Among these solid fuels, peat (30%) shows 2.5-3.8 times higher contribution than the wood (12%) and coal (8%).

In addition to the primary factors, an OOA factor was also resolved and its profile was characterized by a significant contribution from *f44*, which is higher than other primary factors. However, similar to other primary factors, OOA also showed peak concentrations during the evening, indicating OOA was dominated by local sources from residential heating. This is further confirmed by the wind rose which shows higher concentrations of OOA was associated with the lower wind speed from north-northwest direction, consistent with that for the BC polar plots (Fig. S7). The local contribution from heating sources to OOA was probably associated with the condensation of semi-volatile species and/or aging of primary aerosol emitted from biomass burning (Tiitta et al., 2016). However, we cannot rule out the possibility that the smoldering phase of biomass burning was also a potential source of OOA (Weimer et al., 2008). The local and regional sources of OOA will be further discussed below. OOA, on average, accounted for 26% of OA, comparable to HOA (25%) and peat (30%).

### 3.1.2 Carnsore Point

Carnsore Point is located in a rural area on the southeast coast of Ireland, ~150 km south of Dublin. The campaign at Carnsore Point was conducted over the same period as that in Dublin during December 2016 (Fig. 1b). However, the average PM$_1$ concentration was only 2.2 µg m$^{-3}$, ~4 times lower than that in Dublin. OA was the most dominant component, on average accounting for 34% (0.7 µg m$^{-3}$) of PM$_1$, followed by nitrate (22% or 0.5 µg m$^{-3}$), BC (17% or 0.4 µg m$^{-3}$), sulfate (12% or 0.3 µg m$^{-3}$), ammonium (12% or 0.3 µg m$^{-3}$), and chloride (3% or 0.1 µg m$^{-3}$). The fraction of inorganic secondary aerosol (the sum of nitrate, sulfate and ammonium) was 46%, indicating that secondary formation over long-range transport was important





at this rural site. This is consistent with the wind rose of sulfate and nitrate which shows that higher sulfate and nitrate concentrations were associated with wind from the east-southeast direction, pointing to a major source from the UK and/or mainland Europe (Fig. S7). BC is associated with solid fuel combustion and/or biomass combustion. The wind rose of BC shows a major source from northwest areas, pointing to sources from the nearby villages/towns. Therefore, the aerosol

measured at Carnsore Point was impacted by both international long-range transport and the emissions from nearby villages/towns, highly associated with the wind direction.

To investigate the sources of OA, free PMF was firstly conducted on the organic mass spectra. Three factors were identified, including two organic factors (i.e., OOA and SFOA) and one inorganic factor of sea salt (see Fig. S8 and more details in the supplementary). The profile of the sea salt factor was characterized by its fragments of $m/z$ 37, 58, 60, and 83 which were

typical of sea salt fragmentation as found in our previous study where sea salt solution was atomized and directly characterized by AMS (Ovadnevaite et al., 2012). The standard fragmentation table (Allan et al., 2004) in ACSM does not include sea salt which is, however, ubiquitous in the marine environment. As a result, all the sea salt fragmentation ions (i.e., $m/z$ 37, 58, 60, and 83) were included as "organic". Therefore, the true OA were corrected by subtracting the sea salt contribution at Carnsore Point. However, the profile of sea salt factor resolved by free PMF was not "clean" with some interference from other mass

spectral fragments even at higher number-factor solutions (Fig. S8). To better quantify the contribution of sea salt, ME-2 was utilized to constrain its reference profile. A tight constraint ($a$ value of 0.05) was applied because the sea salt factor was not expected to vary significantly.

To evaluate the contribution from different solid fuels to the total SFOA factor, as well as the oil heating factor, the reference profiles of wood, peat, coal, and HOA were constrained with the $a$ value approach using ME-2. The sensitivity test with

varying $a$ values (0-0.5) shows the average relative contribution of the factors did not vary significantly (only by a few percent) within the considered $a$ values (Fig. S9). Figure 4 shows the reference profile and time series of all the factors obtained using the $a$ value of 0.1. The profiles of the HOA, wood, peat, and coal were similar to that found in Dublin as expected because a tight constraint was applied with ME-2 at both locations. HOA, peat, coal, and wood were the primary OA factors. Among these primary factors, peat factor was dominant, accounting for 16% of the total OA mass. The wind rose for the peat factor

shows higher concentrations were associated with wind direction from the northwest at a wind speed of 2-4 m s$^{-1}$, indicating a source region from nearby villages/towns, consistent with that for BC (Fig. S7). HOA, coal, and wood were the minor OA factors, accounting for 4%, 6%, 3% of OA, respectively. In addition to these primary OA factors, an OOA factor was resolved. The profile of OOA featured an $f44$ of 0.29 which was higher than $f44$ of 0.19 for the OOA in Dublin, suggesting the OOA was more oxidized and had undergone more photochemical processing before reaching Carnsore Point. OOA, on average,

accounted for 71% of OA at Carnsore Point, more than twice that in Dublin (26%), again suggesting the importance of secondary formation and/or aging of primary aerosol at this rural site. The wind rose of OOA shows higher concentrations (1-2 µg m$^{-3}$) of OOA was associated with the wind from the east and south-east direction (Fig. S7), pointing to a source from the UK and/or other European countries.





The profile of the sea salt factor was characterized by the prominent signals at *m/z* 37, 58, 60, and 83, corresponding to sea salt fragmentation (Ovadnevaite et al., 2012). Sea salt particles are formed at the sea surface through wave breaking and higher wind speed is usually associated with higher sea salt concentrations (Ovadnevaite et al., 2012). The time series of sea salt showed higher concentration at higher wind speed when the concentration of other factors including peat and OOA factors were very low (Fig. 4). Note that a scaling factor of 11 should be applied to calculate the real sea salt concentration from PMF-ACSM results after comparing with HR-AMS SS concentration (Fig. S10) because the sea salt was not calibrated in the ACSM system. The wind rose for sea salt shows higher concentrations were associated with wind from the south to west direction at a wind speed of 6-12 m s$^{-1}$ (Fig. S11), pointing to a source from the oceanic direction instead of the continental direction. In contrast, the sea salt showed very low concentration (<0.1 μg m$^{-3}$) at low wind speed (<6 m s$^{-1}$), suggesting insufficient sea salt production at low wind speed.

### 3.1.3 Birr

Figure 1c shows the time series of PM$_1$ composition measured by an ACSM and AE-33 in Birr during December 2015. The campaign-averaged PM$_1$ concentration was 4.8 μg m$^{-3}$, ranging from <0.5 μg m$^{-3}$ to 63.0 μg m$^{-3}$ (Table 1). The PM$_1$ chemical composition was dominated by OA, on average, accounting for 62% (2.9 μg m$^{-3}$) of OA, followed by BC (15% or 0.7 μg m$^{-3}$), sulfate (10% or 0.5 μg m$^{-3}$), ammonium (5% or 0.3 μg m$^{-3}$), nitrate (4% or 0.2 μg m$^{-3}$), and chloride (4% or 0.2 μg m$^{-3}$). The time series of OA and BC both showed spike concentrations in the evening, indicating a source from nearby heating activities. The peak OA concentration was 42.1 μg m$^{-3}$, observed in the evening on 13 December 2015, accompanied by a peak concentration of BC (5.8 μg m$^{-3}$), sulfate (8.8 μg m$^{-3}$), ammonium (2.5 μg m$^{-3}$), nitrate (2.1 μg m$^{-3}$), and chloride (1.7 μg m$^{-3}$). Source apportionment of OA using ME-2 showed the spikes were mainly due to solid fuel burning (Fig. S12). On average, solid fuels (the sum of peat, coal, and wood) accounted for 48% (1.2 μg m$^{-3}$) of OA (Table 2). The Peat factor was the most dominant solid fuel factor, on average, accounting for 27% (0.7 μg m$^{-3}$) of OA. During the pollution peak, the contribution from peat increased to 66% (or 22.2 μg m$^{-3}$), highlighting its dominance. Coal and wood factors, on average, accounted for 12% (or 0.3 μg m$^{-3}$) and 9% (or 0.2 μg m$^{-3}$) of the OA, respectively. During the pollution peak, coal and wood factor increased their concentration to 1.6 μg m$^{-3}$ and 1.7 μg m$^{-3}$, respectively. However, the fractions of coal and wood factors were only 4% and 5%, respectively. Similarly, the OOA contribution was higher during the pollution peaks than its average value (3.7 vs. 0.9 μg m$^{-3}$) but its fraction was only 10% compared to the average of 35%. The wind rose for OOA, Peat, and BC all showed their higher concentrations were associated with low wind speed (<5 m s$^{-1}$) from no specific wind directions (Fig. S7), consistent with fact that the measurement site was surrounded by residential households.

### 3.1.4 Mace Head

Figure 1d shows the time series of NR-PM$_1$ components measured by ACSM at Mace Head in January 2013. The average PM$_1$ concentration was 0.7 μg m$^{-3}$, which was the lowest among the four sites, primarily due to the dominant influence of marine air masses at this location. OA dominated the chemical composition of PM$_1$, on average, accounting for 44% (or 0.3 μg m$^{-3}$)





of OA, followed by BC (15%, 0.1 µg m$^{-3}$) and nitrate (15% or 0.1 µg m$^{-3}$). Sulfate (8% or <0.1 µg m$^{-3}$), ammonium (8% or <0.1 µg m$^{-3}$), and chloride (8% or <0.1 µg m$^{-3}$) accounted for the rest 24% of PM$_1$. OA spikes with a concentration of ~5 µg m$^{-3}$ were observed in the evening on 1 January and 5 January 2013. Source apportionment of OA with ME-2 shows these spikes were from the heating source of oil, peat, coal, and wood (Fig S13). Among these primary sources, peat was the greatest

OA factor, on average, accounting for 22% (or 0.06 µg m$^{-3}$) of OA. The contribution from the peat factor increased to 33% (1.3 µg m$^{-3}$) during the pollution peak (Table 2). The wind dependency of the peat factor showed that higher concentration of peat was associated with wind from the east direction at wind speeds < 5 m s$^{-1}$ (Fig. S7), pointing to a source from nearby villages/towns. OOA, on average, accounted for 43% of OA, making it the most dominant OA factor at Mace Head. The wind rose for OOA, sulfate, and nitrate showed that their highest contributions were associated with Easterly wind at wind speed

(>5 m s$^{-1}$). Finally, an inorganic factor of sea salt was resolved at Mace Head. The wind rose for sea salt shows its highest contribution was associated with Westerly wind at high wind speed (>15 m s$^{-1}$), pointing to sea salt production in the Atlantic Ocean during periods with high wind speeds (Fig. S11).

**3.2 Comparison between Dublin and Carnsore Point**

The simultaneous measurements performed at Dublin and Carnsore Point can be compared to gain insight into local versus

regional aerosol sources and to assess their impact on air quality. Primary OA factors including HOA, peat, coal, and wood were directly emitted from their corresponding sources and were mainly associated with local emissions. As shown in Fig. 2, on average, 74% of OA was primary in Dublin while only 29% of OA was primary in Carnsore Point. Thus, the air quality in wintertime Dublin was heavily influenced by local sources while secondary formation and/or long-range transport was more important in Carnsore Point. Secondary organic aerosol and secondary inorganic aerosol (e.g., sulfate, nitrate, and ammonium)

were formed from their precursor gases such as NO$_x$, SO$_2$, and VOCs which could be emitted from sources such as solid fuel burning and traffic. However, secondary aerosol can be formed locally from locally emitted precursor gases or formed over long-range transport.

Figure 5 shows the comparison of sulfate, nitrate, ammonium, and OOA concentration between Dublin and Carnsore Point. Despite the long distance (~150 km) between the two sites, the sulfate time series showed a moderate correlation (R=0.65),

indicating similar sources and/or forming processes (Fig. 6a). However, sulfate also showed some evening spikes (~3 µg m$^{-3}$; e.g., in the evening on 1, 3, and 29 December) in Dublin which was not observed at Carnsore Point. The sulfate spikes in Dublin can thus be attributed to local sources. The formation of sulfate through photochemical reaction pathways was not likely in the evening. In contrast, the high RH (85-100%) during evening hours could enhance sulfate formation via aqueous-phase processing (Sun et al., 2013). As shown in Fig. S14, the evening sulfate spikes simultaneously increased with the

precursor gas SO$_2$, indicating a common source, likely the burning of peat and/or coal. In our previous fingerprinting experiments, sulfate was detected from the direct measurement of peat and coal combustion emissions using an ACSM (Lin et al., 2017). Sulfur, as organic or inorganic compounds in peat or coal, is oxidized to SO$_2$ when burned. Part of SO$_2$ is further oxidized to SO$_3$ by the atomic oxygen formed in flames at a temperature of ~500 °C (Srivastava et al., 2004). The resulting



$SO_3$ rapidly reacts with $H_2O$ to form $H_2SO_4$ at high RH levels. Thereafter, $H_2SO_4$ can form $NH_4SO_4$ through interaction with ammonia in the gas phase or ammonium in particles. Therefore, the observed spikes of sulfate concentrations in Dublin were likely directly emitted from peat and/or coal burning via fast oxidation of $SO_2$ gas to form particle phase $SO_4^{2-}$. After removing the evening spikes, the correlation between sulfate in Dublin and Carnsore Point increased to 0.9 (Fig. 6b) with a slope close

to 1, suggesting sulfate was strongly associated with regional transport during the daytime (8:00-16:00, local time).

The time series of nitrate in Dublin also showed some spikes in the evening, likely due to rapid oxidation of the precursor gases $NO_x$ emitted from solid fuel burning. The nitrate in Carnsore Point, however, most likely originated from regional transport because of a lack of the local sources of precursor gases of $NO_x$. Similarly, OOA in Dublin also showed spikes in concentration during the evening which was associated with local formation, likely from the condensation of semi-volatile

organic species emitted from heating sources. In contrast, OOA in Carnsore Point was most likely of regional origin due to the lack of local sources of its precursor gases as indicated by a relatively low POA fraction. This was consistent with the poor correlation coefficient (R=0.3) of OOA between the two sites (Fig. 6c). Even after removing the evening OOA spikes, the correlation of OOA time series between the two sites did not improve significantly (from 0.3 to 0.49) as compared to the magnitude for sulfate (Fig. 6c and 6d). The overall poor correlation of OOA between the two sites indicates the locally

produced OOA dominated the SOA concentrations in Dublin.

Continental and marine air masses alternately arrived at the measurement sites, bringing aerosols with different composition. As shown in Fig. 5, on 5-6 December 2016, air masses with origins from mainland Europe arrived at the measurement sites. As a result, secondary aerosol such as sulfate, nitrate, ammonium, and OOA concentrations showed a simultaneous increase at the Dublin site and Carnsore Point. For example, nitrate concentration reached a peak of ~6 µg m$^{-3}$ simultaneously at the

two sites. Similarly, OOA concentration peaked at ~4 µg m$^{-3}$ and sulfate concentration at ~1.5 µg m$^{-3}$ simultaneously. The temporal covariation of secondary aerosols at the two sites indicates that the outflow of European aerosol had a great impact, covering an area with a radius of at least 150 km. Averaged over this period, the total PM$_1$ concentration was 8.0-9.0 µg m$^{-3}$ (Table 3) with 75-83% of PM$_1$ being secondary. Among PM$_1$ species, nitrate was the most dominant, accounting, on average, for 29-30% of PM$_1$, followed by OOA, representing 22-26% of PM$_1$.

On 22-27 December 2016, PM$_1$ concentrations were more than 10 times lower than during other periods due to the influence of clean marine air masses (Fig. 5). The average BC concentration was 0.08 µg m$^{-3}$ and the median BC concentration was 0.06 µg m$^{-3}$ at Carnsore Point, indicating a very low impact from anthropogenic aerosol sources. However, the BC concentration at the Dublin site was higher than Carnsore Point, with a mean concentration of 0.40 µg m$^{-3}$ and a median of 0.23 µg m$^{-3}$. The higher BC concentrations in Dublin are attributed to local emissions. Similarly, other non-sea salt PM$_1$ species concentrations

were higher at the Dublin site than Carnsore Point. For example, the average OOA concentration was 0.07 µg m$^{-3}$ at Carnsore Point and 0.65 µg m$^{-3}$ in Dublin. Overall, sea salt dominated the PM$_1$ mass during marine events at Carnsore Point, on average accounting for 65% of the total PM$_1$. As discussed above, this was due to strong winds (>8 m s$^{-1}$) during marine events which resulted in more sea spray.



### 3.3 Spatial distribution and chemical variation of PM$_1$

Dublin was the most polluted area with an average PM$_1$ concentration 2-12 times higher than in the other locations. Birr, a small town in the midlands, was the second most polluted area with an average PM$_1$ concentration about half of that in Dublin. Note that Birr has ~200 times smaller population than Dublin. At the rural coastal sites Carnsore Point and Mace Head, the average PM$_1$ concentration was 4-12 times lower than that in Dublin. However, PM$_1$ spikes due to residential heating emissions from nearby villages were also observed. Overall, the chemical composition of PM$_1$ in Dublin and Birr was very similar, with both locations dominated by carbonaceous aerosol (OA+BC) which accounted for ~80% of PM$_1$. During the pollution events, the fraction of carbonaceous aerosol increased up to 90% of PM$_1$. Therefore, reducing carbonaceous aerosol emissions is important to improve air quality in the cities and towns of Ireland. In contrast, the chemical composition of PM$_1$ at Carnsore Point and Mace Head were similar with inorganic secondary aerosol becoming important which accounted, on average, for 31-46% of PM$_1$.

In agreement with POA being locally emitted rather than regionally transported, urban locations (Dublin and Birr) had higher POA concentrations than the background sites (Mace Head and Carnsore Point). POA, on average, accounted for 65-74% of the total OA in the urban areas while background sites were dominated by OOA which accounted for 43-72% of OA. Among POA factors, solid fuel burning sources were dominant. Consistently, solid fuel contribution was higher in urban areas (48-50% or 1.2-2.2 $\mu$g m$^{-3}$) than the background sites (25-39% or 0.1-0.2 $\mu$g m$^{-3}$) due to proximity to the emission sources. Among these solid fuels, the peat contribution was the most prominent, on average accounting for 16-30% of the total OA (or 0.06-1.3 $\mu$g m$^{-3}$). During the pollution periods, its contribution increased significantly to 32-63% (or 1.3-34.9 $\mu$g m$^{-3}$). These results indicate that, in order to cost-efficiently improve the air quality in urban areas, emissions from solid fuel burning and especially peat burning should be tackled.

The results show that sulfate, nitrate, ammonium, and OOA could originate from local and regional sources. However, we could not exclusively apportion these components into specific sources (e.g., peat or coal burning). The evening spikes of sulfate, nitrate, ammonium, and OOA with other POA factors suggests a similar local source from residential heating. Therefore, the contribution from solid fuel burning could be higher than solely represented by the POA fraction as discussed above. Although OOA shows a higher average contribution (43-71%) at background sites than in the city or the town (26-35%), the absolute OOA concentration (0.1-0.5 $\mu$g m$^{-3}$) at background sites was considerably lower than that in Dublin and Birr (0.9-1.1 $\mu$g m$^{-3}$). This was due to the contribution of locally formed OOA in urban areas while most of OOA at the background sites were regionally formed and transported.

### 4 Conclusion

An ACSM and AE-33 were deployed to characterize the PM$_1$ mass, chemical composition, and sources during winter time across Ireland. The results show that Dublin city was the most polluted area. Birr, a midland town with a population less than 1% of Dublin, had comparable PM$_1$ concentrations and chemical composition. The OA source apportionment results show

that pollution at urban locations was due to local emissions from residential heating with peat, on average, accounting for 27-30% of the total OA mass and even 49-63% during pollution events. Therefore, in order to reduce wintertime particulate air pollution, primary emissions from solid fuel burning, especially peat, should be the primary target of policy regulations. In

contrary, $PM_1$ at Carnsore Point, a regional background site on the southeast coast of Ireland, was dominated by secondary aerosol, with OOA accounting, on average, for 71% of OA. Mace Head, another regional background site on the west coast of Ireland, shows a similar chemical composition to that of Carnsore Point, but the $PM_1$ concentration (0.6 µg m$^{-3}$) was more than 3 times lower due to the longer distance from mainland Europe and greater exposure to the North East Atlantic Ocean. The simultaneous measurements in Dublin and Carnsore point proved that secondary aerosol could be of both local and regional origins. The regional transport of mainland European aerosol featured a simultaneous increase in nitrate, sulfate, ammonium,

and OOA concentration, the sum of which accounted for 79-81% of $PM_1$ while in marine air masses $PM_1$ concentration was more than 10 times lower.

**Data Availability**

All data needed to evaluate the conclusions in the paper are present in the paper and/or the Supplementary Materials. Also, all data used in the study are available from the corresponding author upon request.

**Author Contribution**

JO, DC, MR, MCF, JW, RJH and CO'D conceived and designed the experiments; CL, JO, DC and PB performed the experiments; CL, RJH, JO, WX, PB, TS, DM, SH, JP, and CO'D analysed the data; CL prepared the manuscript with input from all co-authors.

**Competing interests**

The authors declare that they have no conflict of interest.

**Acknowledgments**

This work was supported by EPA-Ireland (AEROSOURCE, 2016-CCRP-MS-31), the National Natural Science Foundation of China (NSFC) under grant no. 91644219 and 41877408, and Chinese Scholarship Council (CSC, no. 201506310020). The authors would also like to acknowledge the contribution of the COST Action CA16109 (COLOSSAL) and MaREI (Center for

marine and Renewable Energy). The team from University College Cork acknowledge support from the Environmental Protection Agency and Department of Environment Community and Local Government in Ireland through the SAPPHIRE



project (2013-EH-MS-15) and from the Irish Research Council (GOIPG/2015/3051). The CNR "Joint Laboratories" Air-Sea Lab project is also acknowledged.

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





**Table 1. Average and peak concentrations of organic aerosol (OA), sulfate (SO₄), nitrate (NO₃), ammonium (NH₄), chloride (Chl), and black carbon (BC), as well as relative contribution (%) to total PM₁ at the four measurement sites across Ireland**

| | Dublin (Dec 2016) | | | | Carnsore Point (Dec 2016) | | | | Birr (Dec 2015) | | | | Mace Head (Jan 2013) | | | |
|---|---|---|---|---|---|---|---|---|---|---|---|---|---|---|---|---|
| | Mean | % | Peak | % | Mean | % | Peak | % | Mean | % | Peak | % | Mean | % | Peak | % |
| OA | 4.9 | 57 | 82.0 | 56 | 0.74 | 34 | 8.6 | 39 | 2.9 | 62 | 42.1 | 67 | 0.32 | 46 | 4.4 | 71 |
| SO₄ | 0.43 | 5 | 2.8 | 2 | 0.27 | 12 | 1.4 | 6 | 0.49 | 10 | 8.8 | 14 | 0.06 | 9 | 0.12 | 2 |
| NO₃ | 0.71 | 8 | 5.6 | 4 | 0.47 | 22 | 4.8 | 22 | 0.22 | 5 | 2.1 | 3 | 0.11 | 16 | 0.19 | 3 |
| NH₄ | 0.33 | 4 | 2.5 | 2 | 0.26 | 12 | 2.2 | 10 | 0.26 | 5 | 2.5 | 4 | 0.06 | 8 | 0.19 | 3 |
| Chl | 0.22 | 3 | 4.3 | 3 | 0.07 | 3 | 0.44 | 2 | 0.17 | 4 | 1.7 | 3 | 0.06 | 5 | 0.31 | 5 |
| BC | 2.0 | 23 | 49.7 | 34 | 0.36 | 17 | 4.5 | 20 | 0.7 | 15 | 5.8 | 9 | 0.11 | 16 | 0.99 | 16 |
| Total | 8.6 | | 146.8 | | 2.2 | | 22.0 | | 4.8 | | 63.0 | | 0.7 | | 6.2 | |

5     **Table 2. Average and peak concentrations of HOA, peat, coal, wood, and oxygenated organic aerosol (OOA), as well as their relative contribution (%) to the total OA mass at the four measurement sites across Ireland**

| | Dublin (Dec 2016) | | | | Carnsore Point (Dec 2016) | | | | Birr (Dec 2015) | | | | Mace Head (Jan 2013) | | | |
|---|---|---|---|---|---|---|---|---|---|---|---|---|---|---|---|---|
| | Mean | % | Peak | % | Mean | % | Peak | % | Mean | % | Peak | % | Mean | % | Peak | % |
| HOA | 1.1 | 24 | 19.2 | 27 | 0.03 | 4 | 0.3 | 3 | 0.4 | 17 | 6.3 | 18 | 0.05 | 18 | 1.2 | 31 |
| Peat | 1.3 | 30 | 34.9 | 49 | 0.1 | 16 | 3.1 | 40 | 0.7 | 27 | 22.2 | 63 | 0.06 | 22 | 1.3 | 32 |
| Coal | 0.3 | 8 | 2.3 | 3 | 0.04 | 6 | 1.6 | 20 | 0.3 | 12 | 1.6 | 4 | 0.03 | 11 | 0.6 | 15 |
| Wood | 0.5 | 12 | 13.5 | 19 | 0.02 | 3 | 0.6 | 8 | 0.2 | 9 | 1.7 | 5 | 0.02 | 6 | 0.5 | 12 |
| OOA | 1.1 | 26 | 2.0 | 3 | 0.5 | 71 | 2.2 | 29 | 0.9 | 35 | 3.7 | 10 | 0.1 | 43 | 0.4 | 9 |
| Total | 4.3 | | 72.0 | | 0.7 | | 7.8 | | 2.6 | | 35.5 | | 0.3 | | 3.9 | |





**Table 3. Average concentrations of secondary organic aerosol (OOA), primary organic aerosol (POA), sulfate (SO$_4$), nitrate (NO$_3$), ammonium (NH$_4$), chloride (Chl), black carbon (BC), and sea salt (SS), as well as their standard deviation (SD) and relative contribution (%) to the total PM$_1$ during continental air mass events on 5-6 December 2016 and marine air mass events on 22-27 December 2016 at the urban background site of Dublin and the rural site of Carnsore Point.**

| | Dublin | | | | | | Carnsore Point | | | | | |
|---|---|---|---|---|---|---|---|---|---|---|---|---|
| | Continental (5-6 Dec) | | | Marine (22-27 Dec) | | | Continental (5-6 Dec) | | | Marine (22-27 Dec) | | |
| | Mean | SD | % | Mean | SD | % | Mean | SD | % | Mean | SD | % |
| OOA | 2.00 | 1.01 | 22 | 0.65 | 0.91 | 33 | 2.05 | 1.10 | 26 | 0.07 | 0.07 | 9 |
| POA[a] | 1.15 | 1.07 | 12 | 0.68 | 1.85 | 33 | 0.48 | 0.38 | 6 | 0.04 | 0.07 | 5 |
| SO$_4$ | 1.00 | 0.39 | 11 | 0.08 | 0.06 | 5 | 0.90 | 0.30 | 11 | 0.05 | 0.03 | 6 |
| NO$_3$ | 2.67 | 2.12 | 30 | 0.11 | 0.21 | 5 | 2.32 | 1.82 | 29 | 0.05 | 0.04 | 6 |
| NH$_4$ | 0.82 | 0.47 | 9 | 0.05 | 0.07 | 2 | 1.00 | 0.72 | 13 | 0.01 | 0.07 | 1 |
| Chl | 0.16 | 0.11 | 2 | 0.04 | 0.06 | 2 | 0.08 | 0.05 | 1 | - | - | - |
| BC | 1.18 | 0.84 | 13 | 0.40 | 0.65 | 19 | 0.87 | 0.04 | 11 | 0.08 | 0.06 | 10 |
| SS | - | - | - | - | - | - | 0.3 | 0.2 | 2 | 0.5 | 0.3 | 63 |
| Total | 9.0 | | | 2.1 | | | 8.0 | | | 0.8 | | |

5   [a] POA is the sum of HOA, peat, coal, and wood factors.





**Figure 1.** Time series of organic aerosol (OA), sulfate (SO₄), nitrate (NO₃), ammonium (NH₄), chloride (Chl) and black carbon (BC) at the urban background site in Dublin (a); the rural site at Carnsore Point (b); the midland town site in Birr (c); and the coastal site at Mace Head (d). The measurements at the Dublin site and Carnsore Point were conducted simultaneously in December 2016. The campaign in Birr was carried out in December 2015 and Mace head was in January 2013. BC, measured by AE-33 with 1 min resolution, was averaged to 30 min to match the time stamp of the ACSM. Pollution periods (P1-P3) and clean periods (C1-C3) in Dublin are marked – see text for further discussion.



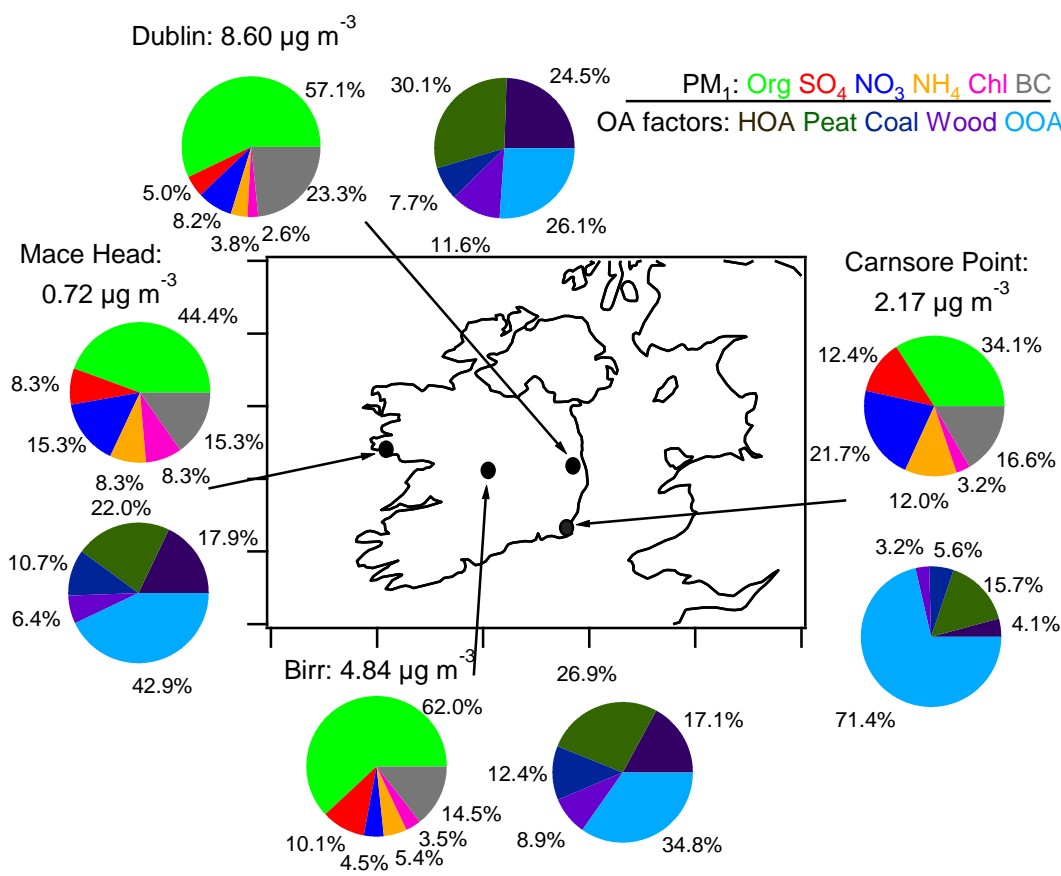

**Figure 2.** Average mass concentration and composition of $PM_1$ and OA factors in Dublin (top), Carnsore Point (right), Birr (bottom), and Mace Head (left).



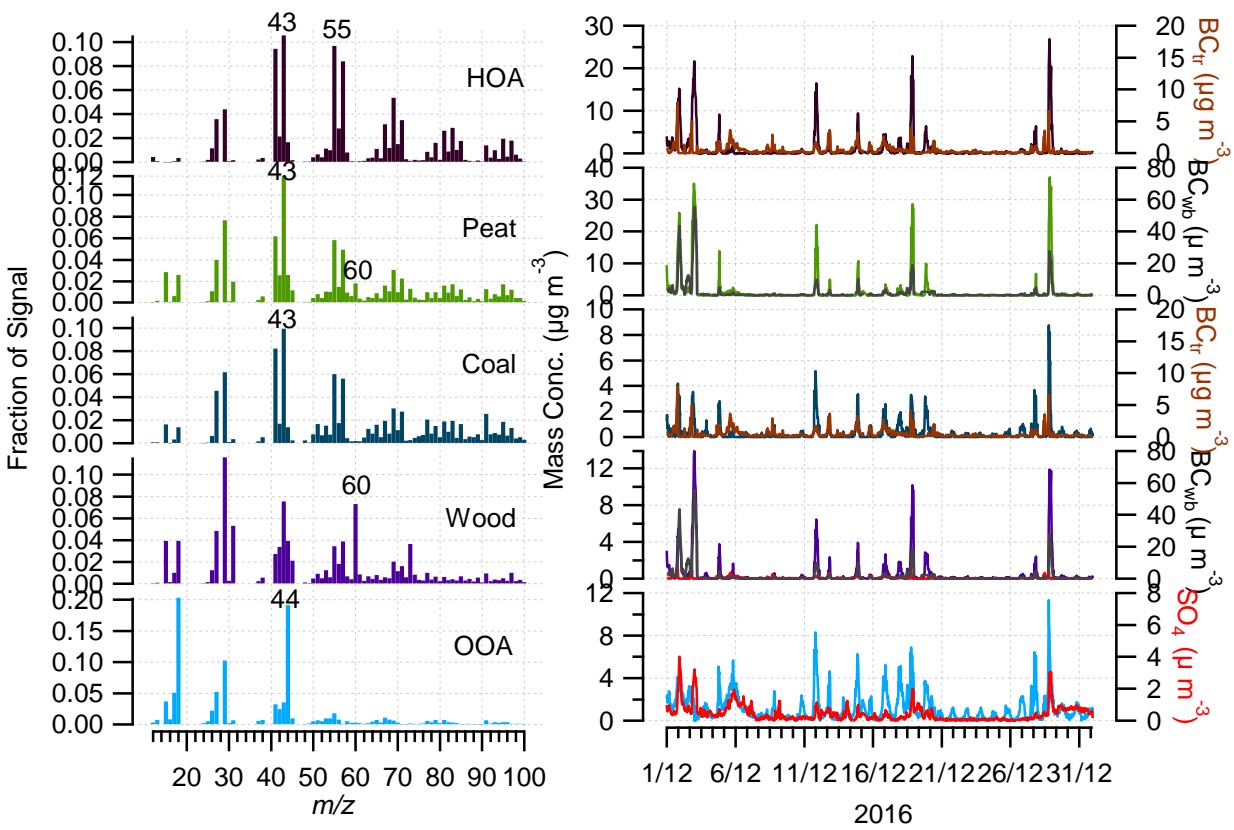

**Figure 3. Profiles and time series of HOA, peat, coal, wood, and oxygenated OA (OOA) at the urban site in Dublin. The time series of BCtr, BCwb, and sulfate (SO4) were also included to support OA source apportionment.**

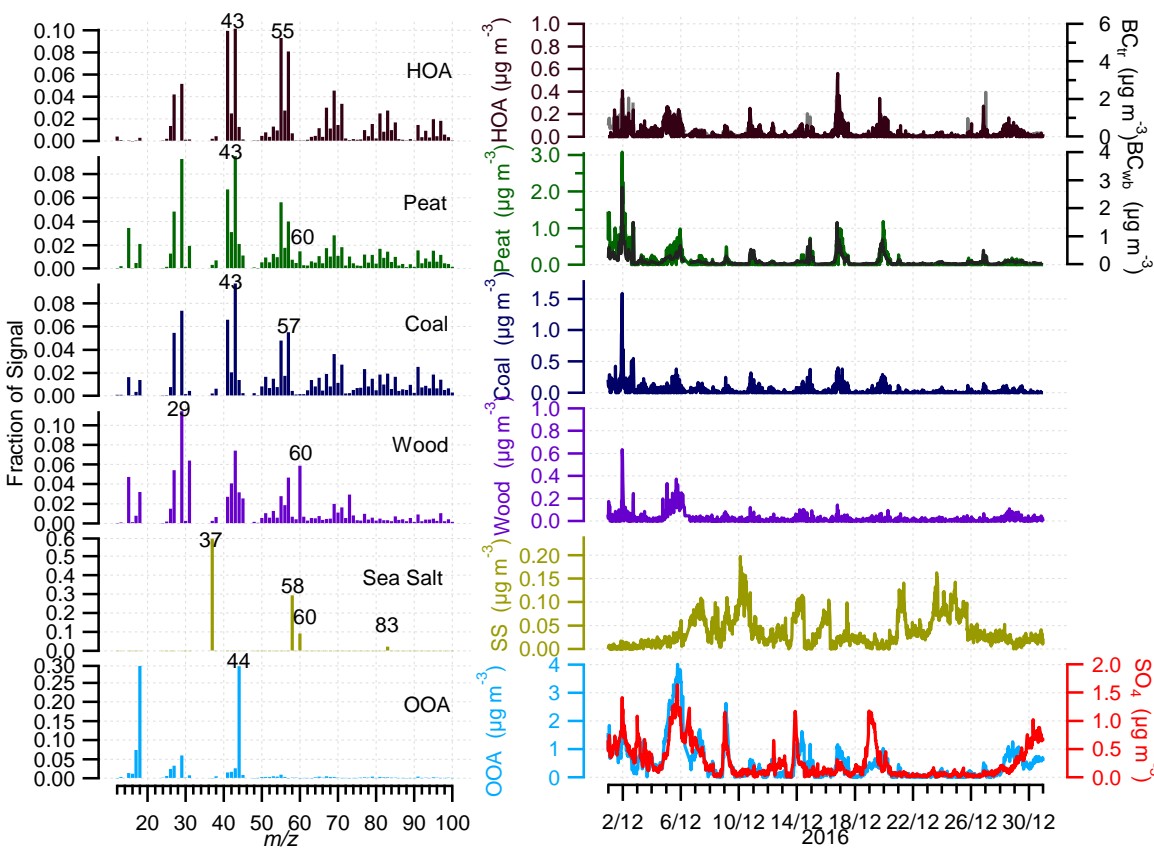

**Figure 4. Profiles and time series of HOA, peat, coal, wood, sea salt, and oxygenated OA (OOA) at the rural site of Carnsore Point. The time series of BCtr, BCwb, and sulfate (SO4) were also included to support OA source apportionment.**



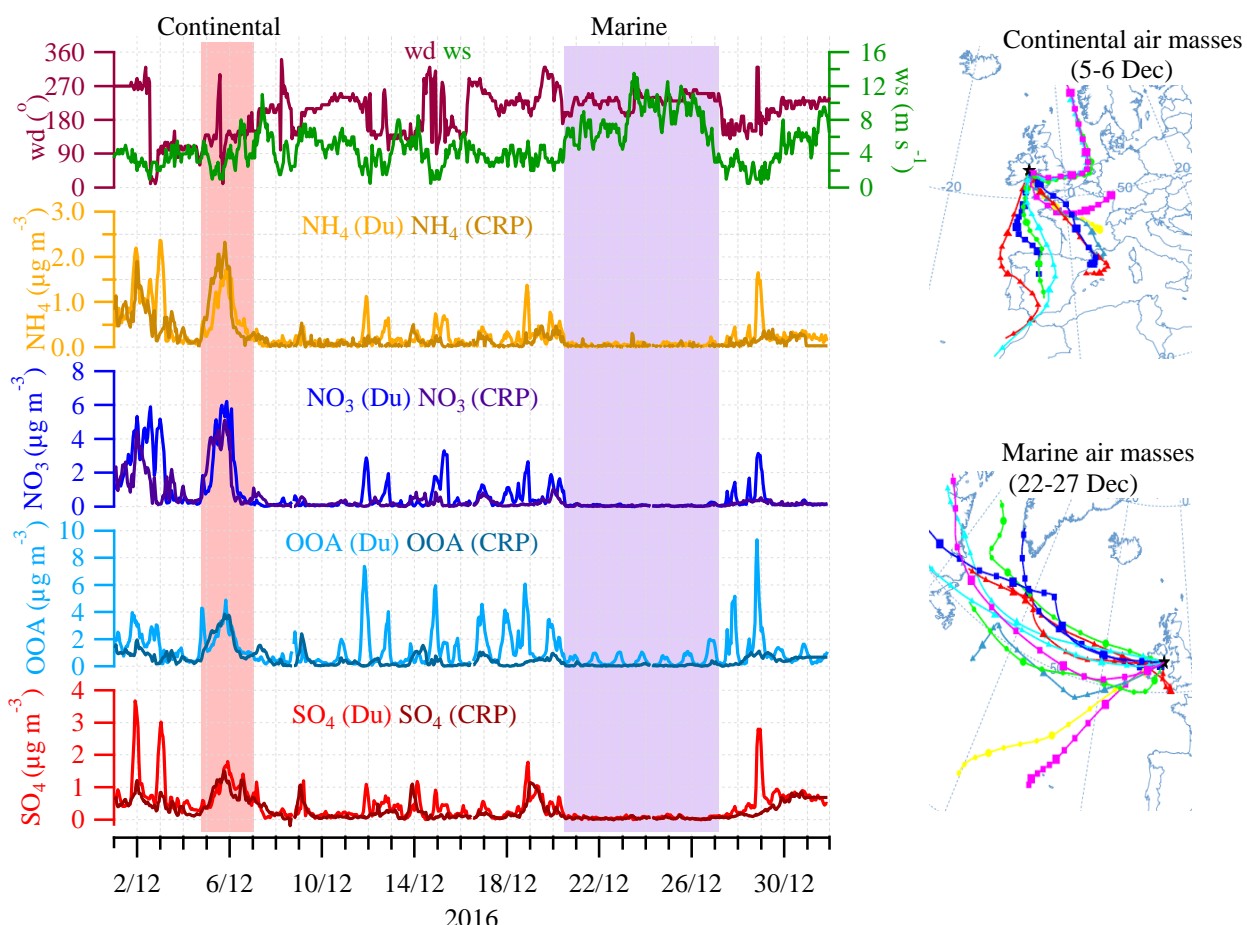

**Figure 5.** Comparison of the time series of sulfate (SO₄), nitrate (NO₃), ammonium (NH₄) and oxygenated organic aerosol (OOA) at the urban site in Dublin (Du) and the rural site in Carnsore Point (CRP). The light red highlighted periods on 5-6 December 2016 were continental air mass periods while the light blue highlighted periods on 22-27 December 2016 were marine air mass periods.

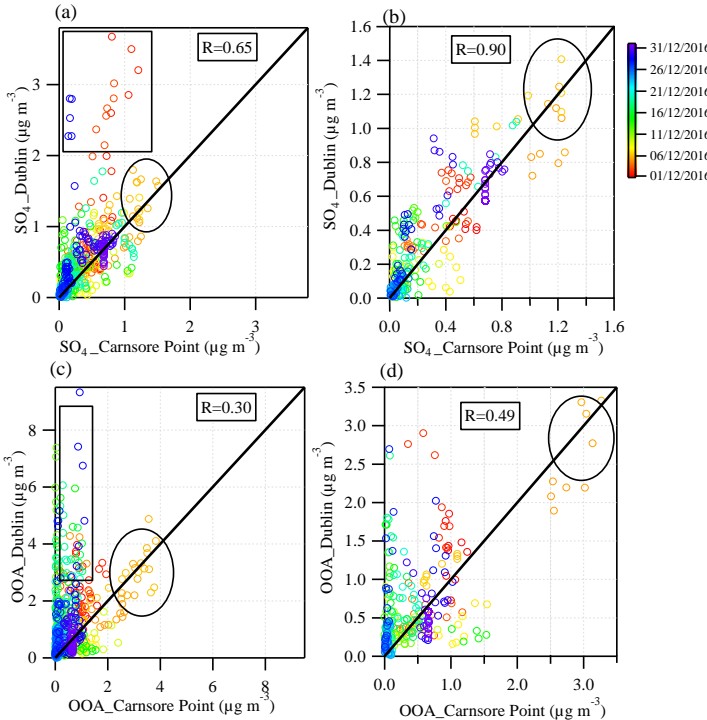

**Figure 6. Scatter plots to compare sulfate (SO₄) and oxygenated organic aerosol (OOA) at the urban site in Dublin (y axis) with the rural site in Carnsore Point (x axis). The rectangle highlights data obtained during periods of local pollution in Dublin. After removing these points, the correlation coefficients increase for sulfate and OOA. The circle highlights points mark the continental events on 5-6 December 2016.**