# Peer review of "Wintertime aerosol dominated by solid fuel burning emissions across Ireland: insight into the spatial and chemical variation of submicron aerosol"

_Atmospheric Chemistry and Physics, 2019_

## Referee Comment (RC1) · Anonymous Referee #1 · 22 Jul 2019

This study analyzes aerosol measurement data collected from four sites in Ireland during different winter periods. Aerosol sources were determined through PMF/ME2 analysis of the ACSM organic aerosol data. From these results, the authors discuss the spatial and chemical variations of PM1 in Ireland. The scope of this work fits well within ACP and the findings could have important implicates for air quality policies and mitigation strategies in Ireland. However, this manuscript has a major problem with its experimental section short of some crucial technical details. As a major focus of this work is source apportionment analysis of the ACSM data, it is imperative that the

manuscript provide thorough discussions on how the results are evaluated and justi-fied. The current discussions are mostly qualitative and sometimes rather subjective. A systematic evaluation of different solutions and the decisions to choose should be pro-vided. Important issues commonly associated with PMF/ME-2 source apportionment results, such as rotational ambiguity, mixing and splitting of factors, and uncertainties in source contribution estimations should also be examined and discussed. Further, rele-vant literatures on the PMF method and its applications in aerosol mass spectrometer data analysis should be cited as well.

Since the measurements were conducted in different years, how does the discussions on aerosol spatial variations affected by the fact that aerosol composition and concen-tration often change considerably from one year to another?

Line 15 on page 3, how was the "urban background site" defined, based on the distance from the city center or some other characteristics of the location?

Section 2.1, mention the distance between the Dublin and the Carnsore Point sites.

Line 10 -12 on page 4, please elaborate on the usage of the Jan 2016 BC data to infer the BC level in 2013, how exactly was it done and under what assumption?

Line 17 – 24 on page 4, the method for determining BCtr and BCwb needs better explanation. The current text is hard to make senses of. Particle absorptions are contributed by both black carbon and brown carbon species. What's the rationale for using absorptions at 470 nm and 880 nm to calculate BCtr and BCwb.

Line 18 -23 on page 4, be specific about the wavelengths used to calculate the AAE values as the number is probably dependent on the pair of wavelength chosen for the calculation.

The HOA discussions on page 6 and 7 need revision. The physical meaning of the HOA factor resolved in Dublin is a bit confusing and some of the discussions are unconvinc-ing and problematic. Dublin is a large city, yet no morning traffic feature is visible in the

HOA diurnal plot. The much larger increase of HOA relative to BC increase at night suggests sources in addition to traffic. The authors jumped to the conclusion of oil heating being a major contributor to nighttime HOA but did not give proper justification. Also, given the large non-traffic influence on the HOA factor, the usage of the HOA/BCtr values to associate HOA with diesel emissions is too speculative. Related texts should be removed.

Page 7, Line 11-12, the sentence "The coal profile featured an f60 of nearly zero which was due to the complete decay of vegetation during coal formation." is difficult to comprehend. Please clarify. Also, an important tracer ion for coal burning OA is C9H7 at m/z 115. What's the behavior of this ion? Is it elevated in the coal burning OA factor?

Page 7, line 24 – 25, this sentence is out of context and the citation of Weimer et al. 2008 is incorrect. The spectra of OOA and BBOA from smoldering burning usually show considerable differences, such as f60 and f73. Weimer et al. mentioned the high m/z44 and little 60 and 73 in the OA spectra of automatic furnace, where the burning condition was unlikely smoldering. Besides, since OA emission is much reduced in the flaming combustion of biomass, gas $CO_2$ contribution could significantly influence the acquired OA spectra. This issue has been discussed extensively in recent papers.

Page 8 line 13, what is identity of the sea salt fragmentation ion at m/z 83?

Figure 3 caption, spell out the differences between BCtr and BCwb.

Figure 5, how far apart are Dublin and Carnsore Point? Is there a basis to assume air pollutants are related between the two locations?

Figure S5, the big drop of r values at a = 0.4 suggests misassignment of the factors.

Figures S3-S5, S8, S9, specify which dataset in the figure caption.

---

## Referee Comment (RC2) · Anonymous Referee #2 · 29 Jul 2019

The study of Lin et al. analyses the PM1 spatial and chemical variation in Ireland using ACSM and AE33 measurements. PM1 spatial variation is very important since a lot of sources are specific to different locations across Europe and are insufficient characterized. Chemical Online measurements offer the opportunity to assess with high accuracy the time evolution of atmospheric aerosol chemical composition. The paper is well-written, making extensive use of the available literature and the results are visualized in an appropriate way. New information is presented in the study related to the main PM1 sources in Ireland during wintertime.

[Figure]

As general remark I recommend that Mace Head measurements and discussion to be threated separately in another study since no data are available in the same time period and BC is assumed to be the constant between 2013 and 2016 wintertime, without a scientific evidence.

Line 15-20 (pp1) Why average concentration for PM 1 in Dublin is comparable with average in Birr ? (the value for Dublin is almost double than Birr).

Line 25-30 (pp3-4) For ACSM should be included all calibration coefficients determined during the campaign measurements, for all sites. Line5-10 (pp 4) Is the CE 1 applied after comparison with SMPS for all sites during the same weather conditions? Did you used SMPS volume concentration? What size range was used for SMPS set-up?

Indeed, the CE did not affect the relative contribution of nonrefractory PM1, but if BC is included in PM1, the relative contribution of BC is dramatically modified, my suggestion is to argue more why CE 1 was chosen.

Chapter 3 (pp 5-6) Is not clear what are the final a values chosen and the correlation values withy BC tracers for final solutions. Please clarify these.

Fig. 5 insufficient explained e.g. the altitude of air masses, number of days used for the model

---

## Author Comment (AC1) · 4 Oct 2019

We are grateful to the referees for their insightful comments which helped to improve the manuscripts substantially. We provided point-by-point responses to the referee's comments below where our responses are in blue.

Referee 1:

This study analyzes aerosol measurement data collected from four sites in Ireland during different winter periods. Aerosol sources were determined through PMF/ME2 analysis of the ACSM organic aerosol data. From these results, the authors discuss the spatial and chemical variations of PM1 in Ireland. The scope of this work fits well within ACP and the findings could have important implicates for air quality policies and mitigation strategies in Ireland. However, this manuscript has a major problem with its experimental section short of some crucial technical details. As a major focus of this work is source apportionment analysis of the ACSM data, it is imperative that the manuscript provide thorough discussions on how the results are evaluated and justified. The current discussions are mostly qualitative and sometimes rather subjective. A systematic evaluation of different solutions and the decisions to choose should be provided. Important issues commonly associated with PMF/ME-2 source apportionment results, such as rotational ambiguity, mixing and splitting of factors, and uncertainties in source contribution estimations should also be examined and discussed. Further, relevant literature on the PMF method and its applications in aerosol mass spectrometer data analysis should be cited as well.

Response: We have added more technical details in the experimental section to show how the source apportionment results were systematically evaluated and justified in the revised manuscript. Also, the rotational ambiguity, mixing, and splitting of factors have also been examined and discussed in the text. More relevant references have also been added.

In Sect. 2.3, it now reads, "Positive Matrix Factorization (PMF) was employed to analyze the contributions of different sources to the measured OA concentrations. The PMF model assumes that measured concentrations at the receptor site can be explained as the linear combination of a source matrix and a contributing matrix (Paatero and Tapper, 1994). Moreover, the PMF model requires all the elements of G and F to be non-negative. The output from the PMF model is a set of factors representing source profiles and source contributions to measured concentrations at the receptor sites. However, the number of factors (i.e., $p$) in PMF is determined by the user and the solutions of the model are not mathematically unique, due to rotational ambiguity.

Unconstrained PMF or free PMF was initially conducted on the OA matrix with a range of solutions and a different number of factors (e.g., from 2 to 8 factors). The solutions were carefully examined and compared with known reference profiles (i.e., mass spectra), derived from literature and/or mass spectra databases (e.g., the AMS spectral database; http://cires1.colorado.edu/jimenez-group/AMSsd/). Moreover, a comparison of factor time series with tracers (e.g., BC) and their diurnal patterns were also important in identifying and evaluating the potential sources.

However, the unconstrained PMF (or free PMF) has difficulties in separating the aerosol sources of temporal covariations. For example, free PMF often fails to separate emissions from different types of solid fuels, which concurrently increase in the evening (Dall'Osto et al., 2013; Lin et al., 2017). Multilinear Engine (ME-2) was utilized to constrain the reference profiles to

direct the source apportionment towards an environmentally meaningful solution (Lanz et al., 2008; Canonaco et al., 2013; Crippa et al., 2014; Reyes-Villegas et al., 2016; Lin et al., 2018). Both free PMF and ME-2 analysis were performed using SourceFinder (SoFi version 6.3, http://www.psi.ch/acsm-stations/me-2), developed by Canonaco et al. (2013). The $a$ value approach of the ME-2 solver was employed to constrain the reference profiles, where the constrained reference profiles were allowed to vary within the scalar value "$a$" (Canonaco et al., 2013). For example, an $a$ value of 0.1 corresponds to 10% variation.

The reference profile of hydrogen-carbon like OA (HOA) was obtained from the literature (Crippa et al., 2013) while the reference profiles of solid fuel factors (i.e., wood, peat, and coal) were taken from our previous fingerprinting experiments conducted in a typical Irish stove with no emission controls (Lin et al., 2017). To explore the solution space, a sensitivity analysis was conducted by varying $a$ values (0-0.5 or 0-50% variation) to evaluate the OA factor contribution at different levels of constraint on the reference factor. At the coastal sites (i.e., Mace Head and Carnsore Point), the reference sea salt profile (Ovadnevaite et al., 2012) was also included to constrain the solution (see more details in Sect. 3.1)."

In Sect. 3.1.1, we have added the following discussion, "High $a$ values (e.g., 0.3-0.5) or a loose constraint led to potential mixing between these heating-related factors especially when their time series showed temporal co-variation (i.e., all showed higher concentration at night). At high $a$ values, the mixing between these factors was evidenced by the sudden drop of correlation coefficient between the time series of the peat factor and $BC_{wb}$ with R dropping from 0.82 at an $a$ value of 0.3 to the R of 0.47 at an $a$ value of 0.4 (Fig. S5) while the correlation between the corresponding profile of e.g., peat also dropped from 0.96 to 0.90 confirming the mixing between peat and other factors (e.g., wood). In contrast, a lower $a$ value (e.g., 0-0.2) reduced mixing and improved the separation by tightly constraining their individual profiles. As shown in Fig. S5, at an $a$ value of 0.1, the time series of peat showed a good correlation with $BC_{wb}$ (R=0.88) while the profile of peat was also tightly correlated with the reference peat profile (R=0.99). Therefore, an $a$ value of 0.1 was chosen as the most optimal ME-2 solution."

Since the measurements were conducted in different years, how does the discussions on aerosol spatial variations affected by the fact that aerosol composition and concentration often change considerably from one year to another?

Response: Actually, the measurements in Dublin (urban) and Carnsore Point (rural) were conducted simultaneously in December 2016, and in Sect 3.2., we have focused on the comparison between Dublin and Carnsore Point to get insights into the spatial variation between urban and rural sites in Ireland. But as the referee noted, the measurements in Birr and Mace Head were carried out in different years (Birr in December 2015 and Mace Head in January 2013). Of course, the aerosol composition and concentration might change in different years depending on the strengths of emission sources. However, the conclusion on the dominance of solid fuel burning in urban areas across Ireland is still robust as our finding is consistent with previous studies conducted in other Irish cities in different years (e.g., Cork in 2008-2009 (Kourtchev et al., 2011; Dall'Osto et al., 2013) and Galway in 2015 (Lin et al., 2017)). In the revised manuscript, we have added the above comments to acknowledge the possible changes in aerosol in different years in Sect. 3.3. "Note that the measurements in Birr and Mace Head were conducted in different years (Birr in 2015 and Mace Head in 2013) than

that in Dublin and Carnsore Point (both in 2016). Therefore, the absolute ratios of the $PM_1$ concentrations between these sites in the same year might vary to a certain degree depending on the strengths of emission sources. However, our finding about the dominance of solid fuel burning in urban areas is consistent with previous studies conducted in other Irish cities in different years (e.g., Cork city in 2008-2009 (Kourtchev et al., 2011; Dall'Osto et al., 2013) and Galway city in 2015 (Lin et al., 2017)). Thus, the conclusion from our study still has significant implications for the air quality policies and mitigation strategies in Ireland, as well as on regional transport for modelling studies."

Line 15 on page 3, how was the "urban background site" defined, based on the distance from the city center or some other characteristics of the location?
Response: The sampling site in Dublin is located in a residential area (i.e., UCD) in South Dublin, ~5km away from the downtown area. The nearest road is ~500 m away, minimizing the influences of direct traffic emissions. Based on these characteristics, the sampling site in UCD, Dublin is defined as the "urban background site". We have added the above information to the revised manuscript in Sect. 2.1.

Section 2.1, mention the distance between the Dublin and the Carnsore Point sites.
Response: The distance between Dublin and Carnsore Point is ~150 km, which is now mentioned in the revised text.

Line 10 -12 on page 4, please elaborate on the usage of the Jan 2016 BC data to infer the BC level in 2013, how exactly was it done and under what assumption?
Response: In the original version, we tried to compare the BC source apportionment with AE-33 across Ireland. However, as both reviewers pointed out, large uncertainty was associated with the usage of the 2016 BC data to infer 2013 BC data in the original version. Therefore, to reduce confusion, we have replaced the AE-33 data in Jan 2016 with MAAP data in Jan 2013 in the revised version.

Line 17 – 24 on page 4, the method for determining BCtr and BCwb needs a better explanation. The current text is hard to make senses of. Particle absorptions are contributed by both black carbon and brown carbon species. What's the rationale for using absorptions at 470 nm and 880 nm to calculate BCtr and BCwb.
Response: We have provided a better explanation in the revised text. It now reads, "BC was apportioned to wood burning-related BC ($BC_{wb}$) and traffic-related BC ($BC_{tr}$) based on their spectral dependence using the Ångström exponent model (Sandradewi et al., 2008; Zotter et al., 2017). Briefly, the spectral dependence of the BC absorption is described by the power law $b_{abs}(\lambda_1)/b_{abs}(\lambda_2) = (\lambda_1/\lambda_2)^{-\alpha}$, where $b_{abs}$ is the aerosol absorption coefficient at the wavelength $\lambda$ while $\alpha$ is the absorption Ångström exponent. BC absorbs light over the entire visible wavelength range with only a weak spectral dependence ($\alpha$ for BC ~ 1). Specifically, traffic emissions contain mostly BC and its absorption is less dependent on the wavelength with $\alpha_{tr}$ of ~1 because traffic emissions basically contain no light-absorbing compounds other than BC (Sandradewi et al., 2008). In contrast, aerosol particles produced from biomass burning contain a substantial amount of light-absorbing organic compounds in addition to BC, which show a

strong increase in absorption in the near-ultraviolet and blue parts of the light spectrum but have no contribution to the absorption at the near-infrared wavelength, resulting in a greater $\alpha_{wb}$ than $\alpha_{tr}$ (Sandradewi et al., 2008; Zotter et al., 2017). Based on this, the measured absorption coefficients at wavelengths 470 nm and 950 nm were used as input to the Ångström exponent model for the apportionment of $BC_{wb}$ and $BC_{tr}$ (Sandradewi et al., 2008). In the original aethalometer two-source model, $\alpha$ (470 - 950 nm) values of 1 and 2 were used for fossil fuel and biomass burning respectively (Sandradewi et al., 2008). However, the most recent evaluation recommends values of $\alpha_{tr}$=0.9 and $\alpha_{wb}$=1.68 (Zotter et al., 2017). These latter $\alpha$ values have been used here."

Line 18 -23 on page 4, be specific about the wavelengths used to calculate the AAE values as the number is probably dependent on the pair of wavelength chosen for the calculation.
Response: The wavelengths of 470 and 950 nm were used to calculate the AAE values. We have added this information to the revised manuscript.

The HOA discussions on page 6 and 7 need revision. The physical meaning of the HOA factor resolved in Dublin is a bit confusing and some of the discussions are unconvincing and problematic. Dublin is a large city, yet no morning traffic feature is visible in the HOA diurnal plot. The much larger increase of HOA relative to BC increase at night suggests sources in addition to traffic. The authors jumped to the conclusion of oil heating being a major contributor to nighttime HOA but did not give proper justification. Also, given the large non-traffic influence on the HOA factor, the usage of the HOA/BCtr values to associate HOA with diesel emissions is too speculative. Related texts should be removed.
Response: Despite Dublin being a large city, the impact from traffic also depends on the distance from the roads, wind speed, wind direction, etc., therefore, it is not very pronounced in the residential measurement location. Actually, to evaluate the impact of traffic emissions on urban air quality, a recent campaign was conducted in Dublin by simultaneously measuring the chemical composition of $PM_1$ at both the kerbside and at the same urban background site in this study. It was found that, while the diurnal cycle of HOA at the kerbside shows typical rush hour peaks, the HOA at the same urban background shows no clear traffic-related patterns. The latter confirms our conclusion that the traffic emissions contribution to HOA at the urban background site is minor (Lin et al., in preparation). Also, as pointed out by the reviewer, the diurnal pattern of HOA features a much larger increase of HOA in the evening when compared to $BC_{tr}$, suggesting other sources (i.e., heating sources) in addition to traffic. We have added the above discussion in the revised manuscript to clarify the potential sources of HOA in Dublin.

Page 7, Line 11-12, the sentence "The coal profile featured an f60 of nearly zero which was due to the complete decay of vegetation during coal formation." is difficult to comprehend. Please clarify. Also, an important tracer ion for coal burning OA is C9H7 at m/z 115. What's the behavior of this ion? Is it elevated in the coal burning OA factor?
Response: Clarified. In the mass spectral signatures for the wood and peat OA factors, the contribution of the signal at *m/z* 60 (that is, *f60*) and 73 (*f73*) to the total organics are associated with fragmentation of levoglucosan. *f60* and *f73* are therefore often regarded as tracers for biomass burning emissions (Alfarra et al., 2007; Cubison et al., 2011; Dall'Osto et al., 2013).

In contrast, the mass spectral signature for the coal OA factor does not have any contribution from *m/z 60* due to the lack of levoglucosan in this fossil fuel (Zhang et al., 2008). We have clarified this in the revised manuscript.

For the OA source apportionment, the input matrix for PMF only covered m/z 12 to 100 because ions outside of this range had poor signal to noise ratios. Therefore, m/z 115 was not included in the PMF analysis to reduce the OA source apportionment uncertainty. In addition to m/z 115, other PAH-derived ions e.g., m/z 77 and 91, also show higher *f77* and *f91* in the coal burning OA factor when compared to wood and peat (Fig. 3 and Fig. 4), consistent with our previous fingerprinting studies (Lin et al., 2017).

Page 7, line 24 – 25, this sentence is out of context and the citation of Weimer et al. 2008 is incorrect. The spectra of OOA and BBOA from smoldering burning usually show considerable differences, such as f60 and f73. Weimer et al. mentioned the high m/z44 and little 60 and 73 in the OA spectra of automatic furnace, where the burning condition was unlikely smoldering. Besides, since OA emission is much reduced in the flaming combustion of biomass, gas CO2 contribution could significantly influence the acquired OA spectra. This issue has been discussed extensively in recent papers.

Response: We accept that the study by Weimer et al. (2008) was conducted with a different type of stove and the citation is inappropriate as pointed out by the referee. To reduce the confusion, we have removed this sentence and the relevant citation.

Page 8 line 13, what is identity of the sea salt fragmentation ion at m/z 83?

Response: The identity of the sea salt fragmentation ion at m/z 83 is $^{23}Na_2^{37}Cl^+$. In the revised text, we have added the fragments for all sea salt-related fragmentation ion "…*m/z* 37 ($^{37}Cl^+$), 58 ($^{23}Na^{35}Cl^+$), 60 ($^{23}Na^{37}Cl^+$), and 83 ($^{23}Na_2^{37}Cl^+$)… Note that other *m/z's* that belong to sea salt, like *m/z* 23 ($^{23}Na^+$) or 81 ($^{23}Na_2^{35}Cl^+$), do not appear in the OA factor profile as they mainly belong to inorganic ions which were not included in the OM matrix for PMF analysis."

Figure 3 caption, spell out the differences between BCtr and BCwb.

Response: Corrected. In the text, "…BC from traffic ($BC_{tr}$), BC from wood burning ($BC_{wb}$), and…"

Figure 5, how far apart are Dublin and Carnsore Point? Is there a basis to assume air pollutants are related between the two locations?

Response: The distance between Dublin and Carnsore Point is ~150 km. Sulfate is usually regarded as a regional pollutant, and the good correlation of sulfate between the two sites confirmed its regional nature in our study (see Sect 3.2 and Fig. 6). In particular, the simultaneous increase in secondary aerosols during the continental air masses from 5-6 December 2016 and the simultaneous decrease in the relatively clean marine air masses during 22-27 December 2017 suggest that air pollutants between the two sites were related in these cases.

Figure S5, the big drop of r values at a = 0.4 suggests misassignment of the factors. Figures S3-

S5, S8, S9, specify which dataset in the figure caption.

Response: Through the examination of the profile at *a*=0.4 (Figure R1), the peat and wood OA factor were mixed but still retained the features of their respective reference profiles in the resulting solution. This is not surprising because all solid fuel factors featured similar temporal variation with higher concentrations during the night, and large *a* values (i.e., 0.4) led to potential mixing between these factors. Therefore, to reduce the mixing between factors, small *a* values (i.e., a <0.2) were preferred when constraining these reference profiles.

We have specified the dataset in the figure caption for Figs. S3-S5, S8 and S9.

[Figure]

Figure R1. the profiles of HOA, peat, coal, wood, and OOA at *a* value of 0.4 in Dublin.

References:

[revised manuscript text omitted]

---

## Author Comment (AC2) · 4 Oct 2019

We are grateful to the referees for their insightful comments which helped to improve the manuscripts substantially. We provided point-by-point responses to the referee's comments below where our responses are in blue.

Referee 2:

The study of Lin et al. analyses the PM1 spatial and chemical variation in Ireland using ACSM and AE33 measurements. PM1 spatial variation is very important since a lot of sources are specific to different locations across Europe and are insufficient characterized. Chemical Online measurements offer the opportunity to assess with high accuracy the time evolution of atmospheric aerosol chemical composition. The paper is well-written, making extensive use of the available literature and the results are visualized in an appropriate way. New information is presented in the study related to the main PM1 sources in Ireland during wintertime.

Response: We thank the referees for the positive feedback. Below, we provide point-to-point response to the referee's comments.

As general remark I recommend that Mace Head measurements and discussion to be threated separately in another study since no data are available in the same time period and BC is assumed to be the constant between 2013 and 2016 wintertime, without scientific evidence.

Response: We agree that the assumption for the constant BC concentration between 2013 and 2016 lacks scientific evidence. In the original version, we tried to compare the BC source apportionment with AE-33 across Ireland. However, as both reviewers pointed out, large uncertainty was associated with the usage of the 2016 BC data to infer 2013 BC data in the original version. Therefore, to reduce confusion, we have replaced the AE-33 data in Jan 2016 with MAAP data in Jan 2013 in the revised version. Also in the revised version, we tend to keep the Mace Head data as part of this comparison study rather than being treated in another study because we believe the comparison of the OA source apportionment between Carnsore Point at the east coast and Mace Head at the west coast of Ireland are very insightful for the characterization of the spatial variation of NR-PM$_1$ and OA sources despite their measurements in different years.

Line 15-20 (pp1) Why average concentration for PM 1 in Dublin is comparable with average in Birr? (the value for Dublin is almost double than Birr).

Response: Corrected. It now reads, "Birr, a small town in the midlands area of Ireland with a population <1% of that in Dublin, showed an average PM$_1$ concentration (4.8 μg m$^{-3}$, ranging from <0.5 to 63.0 μg m$^{-3}$ in December 2015) around half of that (56%) in Dublin"

Line 25-30 (pp3-4) For ACSM should be included all calibration coefficients determined during the campaign measurements, for all sites.

Response: We have added a supplementary Table for all the calibration coefficients for all sites in the revised text. In the text, "…Ionization efficiencies (IEs) and relative ionization efficiencies (RIEs) for sulfate and ammonium were determined through the calibration with ammonium nitrate and ammonium sulfate following the procedure described by Ng et al. (2011), and the IEs and RIEs at each site were provided in Table S1…"

Table S1. Ionization efficiencies (IE) and relative ionization efficiencies (RIE) obtained through the calibration with ammonium nitrate and ammonium sulfate.

|  | Dublin | Carnsore Point | Birr | Mace Head |
|---|---|---|---|---|
| IE | 3.24e-11 | 3.28e-11 | 3.24e-11 | 4.47e-11 |
| RIE (sulfate) | 0.96 | 0.74 | 0.96 | 0.56 |
| RIE (ammonium) | 7.58 | 6.74 | 7.58 | 4.76 |

Line5-10 (pp 4) Is the CE 1 applied after comparison with SMPS for all sites during the same weather conditions?

Response: Actually, only the ACSM deployed in Dublin was compared with the co-located SMPS during the winter of 2016. Because the same ACSM instrument was also deployed in Birr (December 2015) and Mace Head (January 2013) during wintertime with similar weather conditions, CE of 1 was also applied to the ACSM dataset at these two sites. For the ACSM deployed at Carnsore Point, the same magnitude of increase in the concentrations of PM1 as that Dublin during 5-6 December 2016 (Fig. 5 and 6) confirmed the application of CE=1 was also physically meaningful at Carnsore Point.

We have clarified this in the revised text, it now reads, "For all ACSM measurements, a collection efficiency (CE) of 1 was applied for all the measured species. This CE was validated against a collocated scanning mobility particle sizer (SMPS) which shows the sum of the calculated ACSM volume and black carbon (BC) volume correlated well (r = 0.96 and slope = ~1) with the SMPS volume (size ranged from 14.6 nm to 685.4 nm) at the sampling site in Dublin during the winter of 2016 (Lin et al., 2018). Note that the same ACSM was also deployed in Birr during December 2015 and Mace Head during January 2013 under similar weather conditions and thus a CE of 1 was also applied for the datasets at these two sites. For the ACSM at Carnsore Point, the similar magnitude of increase in $PM_1$ in continental air masses (See Sect. 3.2) confirmed that the application of CE of 1 for the Carnsore Point dataset was physically meaningful. Also, note that a CE of 1 provided a lower limit for all ACSM-measured mass concentration."

Did you used SMPS volume concentration? What size range was used for SMPS set-up? Indeed, the CE did not affect the relative contribution of nonrefractory PM1, but if BC is included in PM1, the relative contribution of BC is dramatically modified, my suggestion is to argue more why CE 1 was chosen.

Response: Yes, the SMPS volume concentration was used with the size ranging from 14.6 nm to 685.4 nm. Also, see the response for the previous comment.

Chapter 3 (pp 5-6) Is not clear what are the final $a$ values chosen and the correlation values with BC tracers for final solutions. Please clarify these.

Response: We have added more details on how the final $a$ values were chosen and added correlation values for the BC tracers for the final solutions. In the revised text, it now reads, "High $a$ values (e.g., 0.3-0.5) or a loose constraint led to potential mixing between these heating-related factors especially when their time series showed temporal co-variation (i.e., all showed higher concentration at night). At high $a$ values, the mixing between these factors was evidenced by the sudden drop of correlation coefficient between the time series of the peat factor and $BC_{wb}$ with R dropping from 0.82 at an $a$ value of 0.3 to the R of 0.47 at an $a$ value of 0.4 (Fig. S5) while the correlation between the corresponding profile of e.g., peat also

dropped from 0.96 to 0.90 confirming the mixing between peat and other factors (e.g., wood). In contrast, a lower *a* value (e.g., 0-0.2) reduced mixing and improved the separation by tightly constraining their individual profiles. As shown in Fig. S5, at an *a* value of 0.1, the time series of peat showed a good correlation with BCwb (R=0.88) while the profile of peat was also tightly correlated with the reference peat profile (R=0.99). Therefore, an *a* value of 0.1 was chosen as the most optimal ME-2 solution."

Fig. 5 insufficient explained e.g. the altitude of air masses, number of days used for the model
Response: We have now added more information regarding the HYSPLIT model. It now reads, "…The back trajectories (BTs) on the right panel was calculated using the Hybrid Single-Particle Lagrangian Integrated Trajectory (HYSPLIT; Stein et al. (2015)). The BTs were calculated for an arrival height of 500 m at the length of 72 h, and were every 6 h during continental air masses during 5-6 December (top right) and every 12 h during marine air masses during 22-27 December (bottom right)."

References:
Lin, C., Huang, R.-J., Ceburnis, D., Buckley, P., Preissler, J., Wenger, J., Rinaldi, M., Facchini, M. C., O'Dowd, C., and Ovadnevaite, J.: Extreme air pollution from residential solid fuel burning, Nature Sustain., 2018.
Ng, N. L., Herndon, S. C., Trimborn, A., Canagaratna, M. R., Croteau, P. L., Onasch, T. B., Sueper, D., Worsnop, D. R., Zhang, Q., Sun, Y. L., and Jayne, J. T.: An Aerosol Chemical Speciation Monitor (ACSM) for routine monitoring of the composition and mass concentrations of ambient aerosol, Aerosol Sci. Technol., 45, 780-794, 2011.
Stein, A. F., Draxler, R. R., Rolph, G. D., Stunder, B. J. B., Cohen, M. D., and Ngan, F.: NOAA's HYSPLIT Atmospheric Transport and Dispersion Modeling System, Bull. Am. Meteorol. Soc., 96, 2059-2077, 10.1175/bams-d-14-00110.1, 2015.